# Effects of RF Electric Currents on Hair Follicle Growth and Differentiation: A Possible Treatment for Alopecia

**DOI:** 10.3390/ijms25147865

**Published:** 2024-07-18

**Authors:** María Antonia Martínez-Pascual, Silvia Sacristán, Elena Toledano-Macías, Pablo Naranjo, María Luisa Hernández-Bule

**Affiliations:** 1Photobiology and Bioelectromagnetic Laboratory, Instituto Ramón y Cajal de Investigación Sanitaria (IRYCIS), Hospital Ramón y Cajal, Crta. Colmenar Viejo, km. 9.100, 28034 Madrid, Spain; m.antonia.martinez@hrc.es (M.A.M.-P.); elena.toledano@hrc.es (E.T.-M.); 2Aptamer Group, Histology Laboratory, Instituto Ramón y Cajal de Investigación Sanitaria (IRYCIS), Hospital Ramón y Cajal, Crta. Colmenar Viejo, km. 9.100, 28034 Madrid, Spain; silvia.sacristan@hrc.es; 3Elite Laser Clinic, C/de Orense, 56, 28020 Madrid, Spain; naranjopablo@hotmail.com

**Keywords:** androgenic alopecia, hair follicle, melanoblast, radiofrequency therapy, CRET

## Abstract

Androgenic alopecia (AGA) is the most common type of alopecia and its treatments involve drugs that have various adverse effects and are not completely effective. Radiofrequency-based therapies (RF) are an alternative for AGA treatment. Although there is increasing clinical evidence of the effectiveness of RF for alopecia, its effects at the tissue and cellular level have not been studied in detail. The objective of this study was to analyze ex vivo the potential effect of RF currents used in capacitive resistive electrical transfer (CRET) therapy on AGA. Hair follicles (HFs) were donated by patients with AGA and treated with CRET. AGA-HFs were exposed in vitro to intermittent 448 kHz electric current in subthermal conditions. Cell proliferation (Ki67), apoptosis (TUNEL assay), differentiation (β-catenin), integrity (collagen and MMP9), thickness of the epidermis surrounding HF, proportion of bulge cells and melanoblasts in AGA-HF were analyzed by immunohistochemistry. CRET increased proliferation and decreased death of different populations of AGA-HF cells. In addition, the melanoblasts increased in bulge and the epidermis surrounding the hair follicle thickened. These results support the effectiveness of RF-based therapies for the treatment of alopecia. However, clinical trials are necessary to know the true effectiveness of CRET therapy and other RF therapies for AGA treatment.

## 1. Introduction

Hair is a characteristic feature of mammals and performs a variety of functions, such as thermal insulation, physical protection, camouflage, social interaction, and sensory perception [1]. At the base of each hair resides a mini organ called the hair follicle (HF) invaginated below the surface of the skin and formed by different structures such as the root sheath (internal and external), the hair shaft, the sebaceous gland, the bulge, the dermal sheath and the dermal papilla. These structures are composed of more than twenty different cell populations, including stem cells, keratinocytes, melanocytes, neurons, endothelial cells, derivatives of mast cell precursors and immune cells. These populations appear and disappear, or undergo substantial changes, depending on the phase of the hair growth cycle or the hormonal stimulus present at that time [1].

Hair growth is organized into three cyclic phases of variable duration. On the scalp, the anagen or active growth phase requires 2–7 years to complete, the catagen or involution phase, 2–3 weeks, and the telogen or rest phase, approximately 100 days. After telogen, the lower part of the follicle regenerates, which marks the beginning of a new anagen [2]. Hair loss or thinning is a disorder associated with the deregulation of this growth cycle, resulting from factors such as aging, alterations in hormonal secretion, nutritional imbalances, social stress and genetic factors [3,4]. Although not life-threatening, hair loss can significantly decrease psychological well-being and quality of life. AGA is the most common type of alopecia and is characterized by progressive miniaturization of the hair follicle. Although the pathological mechanism of AGA is still unclear, it has been suggested that the HF bulge plays an important role in its pathogenesis. Since the bulge is a niche of multipotent stem cells necessary for regeneration of the follicle, sebaceous gland and epidermis, it has been proposed that the development of AGA could be due to alterations in the conversion of these bulge stem cells to progenitor cells [5].

Currently, the most used treatment to stop hair loss and promote hair growth includes the drugs minoxidil, finasteride, dutasteride and Janus kinase (JAK) inhibitors. However, these compounds are not curative, their efficacies are transient as hair loss may resume after discontinuation of use [6,7], and they have a number of adverse side effects [8]. Thus, oral minoxidil can cause postural hypotension, fluid retention and hypertrichosis. Finasteride and dutasteride have been linked to persistent sexual abnormalities, anxiety and depression [9] and finasteride has a potential teratogenic effect, so its use with caution is recommended in patients with hepatic impairment [10]. Other drugs that are increasing in popularity among patients with alopecia areata are JAK inhibitors, but common complications include upper respiratory tract infections, elevated enzymes, and headaches [9]. Pharmacological alternatives based on compounds of plant origin have been described for patients who experience side effects with usual drugs, but have limited effectiveness [11]. Biological therapy based on platelet-rich plasma (PRP) is also currently used. PRP is an emerging therapy that uses an autologous preparation consisting of growth factors, cytokines and concentrated platelets, which stimulates hair follicle repair and regeneration. This strategy is relatively safe with minimal adverse effects including temporary and tolerable pain. Microneedles can be used in combination with PRP to promote or facilitate its penetration or enhance its action [9]. Other therapeutic options for the treatment of alopecia are cryotherapy and intralesional corticosteroid (ILCS) injection. According to the recent evidence, both modalities are safe and effective in the treatment of alopecia areata, although cryotherapy is more effective and less painful than ILCS injection, which is why it is recommended as a treatment option, especially in pediatric patients with alopecia areata or when there is a contraindication to ILCS injection [12]. Despite the effectiveness of these therapies, the studies carried out are scarce, so more research with larger samples and similar experimental procedures is necessary to evaluate the clinical effectiveness of these therapies [13].

Other alternative therapies to pharmacological or biological therapies for the treatment of AGA are physical therapies. Among these therapies, lasers have been shown to be capable of inducing hair growth [14]. Fractional lasers create very light localized stimulation that induces a wound healing response to promote hair regrowth [15,16]. Other therapies based on photobiomodulation (PBM) and using LEDs (light-emitting diodes) or a low-level laser (LLLT) have also been proposed for the treatment of alopecia in recent years [17,18]. Thus, wavelengths in the range from 630 nm to 800 nm have been used for the treatment of androgenic alopecia, frontal fibrosing alopecia, alopecia areata and lichen planopillaris. The effect induced by photobiomodulation is a prolongation of the anagen phase of the hair cycle [19]. These therapies have proven to be useful effectively, safely and with relatively mild adverse reactions and, although they are efficient, they have some limitations in terms of application. For example, lasers are applied locally and require many exposures and time to treat a large area. In addition, the length of the hair is a problem for the application of the laser because it absorbs the laser energy, blocking its passage to the skin, so short hair is necessary, which is not well accepted by female patients. Furthermore, light therapies are limited by skin color, and darker skin requires very long treatments [20].

In addition to physical therapies based on light, those that use electromagnetic fields or electric currents are another current option for the treatment of alopecia. In particular, radiofrequency (RF) currents are high-frequency alternating currents whose main property is to increase tissue temperature (hyperthermia). The effects of hyperthermia are directly related to the type of tissue (tissue resistivity to the passage of current) and the exposure time [21,22,23]. Radiofrequency can be used for ablative and non-ablative applications. Among its non-ablative effects are the induction of fibroblast proliferation [24,25,26], activation of neocollagenesis [24,26], improvement of blood circulation [27], neoangiogenesis [24], collagen remodeling and skin tightening [28,29]. Furthermore, its subthermal application (treatments that do not increase temperatures above 37 °C in the target tissues) induces anti-inflammatory and antiedematous effects [30]. Unlike other physical therapies, RF technologies can be applied to any type of skin using non-invasive or minimally invasive procedures and with minimal risk of complications or side effects [31]. Therefore, RF devices are increasingly used in cosmetic dermatology and have been successfully applied for aesthetic treatments such as sagging skin, wrinkles, body/skin rejuvenation, treatments for facial expression lines, recent and late fibrosis, scars and adhesions, cellulite, localized fat and also for alopecia [32]. It should be noted that the radiofrequency used in cosmetic dermatology for hair follicle growth is different from that used for hair removal. Both are high-frequency alternating currents that produce localized heat, but while in the case of hair follicle growth, RF is used for moderate heating of the tissues (thermotherapy), hair removal uses RF to cause a thermal burn (thermocoagulation). To promote thermocoagulation, a small electrode is used, such as a needle, which can concentrate large amounts of heat in the area, causing the destruction of the hair follicle [33]. Clinical evidence supports the use of bipolar and monopolar RF devices for hair removal, while fractional RF appears to be an emerging technology for hair growth [34]. Therefore, the parameters of the devices, the mode of application and their mechanisms of action of radiofrequency are different.

One of the RF therapies that applies this type of electrical stimulation is CRET. This therapy applies 448 kHz currents to induce electrical or electrothermal stimulation in target tissues. Although there is increasing clinical evidence of the effectiveness of RF for alopecia, its effects at the tissue and cellular level have not been studied in detail. The objective of this study was to analyze ex vivo the potential effect of subthermal currents used in CRET therapy on hair follicles, donated by patients with AGA (AGA-HF). Two types of CRET signals were studied: a standard 448 kHz sinusoidal signal (CRET-Std) and a 448 kHz sinusoidal signal modulated by 40% in amplitude at 20 kHz (CRET-Mod). The effect of CRET on the growth, death and cell differentiation of AGA-HF was analyzed.

## 2. Results

### 2.1. Effect of CRET on Proliferation of AGA-HF Cells

Labeling of the Ki67 protein reveals cells in a proliferative state. In this study, the proliferation of cells in different areas of the follicle was analyzed using this Ki67 marker. Although the results obtained (Figure 1) showed interpersonal variations depending on the patient, increases were observed with both CRET signals (modulated and standard) in the proliferation rate compared to controls. The effect was more consistent, generalized and significant when the samples were treated with the standard signal than with the modulated one, since CRET-Mod only increased the rate of Ki67+ cells in the inner root sheath (IRS) and outer root sheath (ORS). In contrast, CRET-Std showed increases in Ki67+ cells in the matrix, bulge cells, and epidermis. Given that AGA-HF has a low proliferation rate, this increase in the proliferation of these cells would indicate an improvement in the follicle.

### 2.2. Effect of CRET on Apoptosis of AGA-HF Cells

The cell death rate in HF was analyzed using the TUNEL assay. Alopecia is characterized by a miniaturization of the follicle due to an increase in the apoptosis of its cells [35]. The results (Figure 2) showed an effect dependent on the type of signal used. Thus, compared to the control group, treatment with the standard signal caused a significant decrease (32.3%) in the rate of TUNEL+ cells in the sheath area (IRS+ORS). No changes were observed in the rest of the HF areas analyzed. When the HF were treated with the modulated signal, no significant differences were found in the rate of TUNEL+ cells in any of the areas of the treated follicles, compared to their controls. Therefore, a reduction in the apoptotic rate in the different parts of the follicle would promote the strengthening of HF, making it difficult to lose it.

### 2.3. Effect of CRET on Epidermal Depth of AGA-HF

In this study, the depth of the epidermis was assessed, which in our samples corresponds to the thickness of the epidermal layer surrounding the follicle. Treatment of the follicles with the standard signal caused a significant increase in the depth of the epidermis, compared to untreated AGA-HF. No changes were observed when the follicles were treated with the modulated signal over AGA-HF control (Figure 3). This increase in the thickness of this layer would reinforce the stability of the AGA-HF, preventing hair loss.

### 2.4. Effect of CRET on AGA-HF Integrity: Analysis of Collagen Production

Potential changes in collagen production in AGA-HF induced by CRET were also analyzed. A degradation of the extracellular matrix leads to a weakening of the structural support of the hair follicle, which can contribute to miniaturization and eventual hair loss. Given the relevance of collagen in the HF matrix, an assessment of the total amount of collagen was performed using Masson’s trichrome staining and subsequent colorimetric image analysis of the dermis. This staining did not reveal significant changes in the amount of color in the samples treated with the standard or modulated signal, compared to the respective controls. Additionally, the expression of collagen type I (Col I) was analyzed by immunohistochemistry. This study showed an increase in Col I in AGA-HF treated with the modulated signal compared to controls, while the standard signal did not modify its expression (Figure 4).

### 2.5. Effect of CRET on AGA-HF Integrity: Analysis of Metalloproteinase 9 (MMP9) Expression

MMPs are involved in the integrity of follicles and the presence of MMP-9+ cells has been described in the sebaceous gland during the anagen and catagen phase, and in the internal root sheath during the catagen phase. Furthermore, the increase in MMP9 activity has been related to hair bulb involution, both in vitro and in vivo [36]. In the follicles analyzed in the present study, the expression of MMP9 was studied in the periglandular areas and in Henle’s layer (located in the inner root sheath). The results (Figure 5) showed significant decreases in both areas when the follicles were treated with the standard signal, compared to their control. No significant changes were observed compared to controls when the signal used was modulated.

### 2.6. Effect of CRET on Follicle Differentiation Factors: Analysis of β-Catenin Expression in AGA-HF

One of the main differentiation pathways involved in hair follicle development is the Wnt/β-catenin cell signaling pathway. In the present study, the expression of β-catenin in the different areas of the AGA-HF was analyzed. The treatment of AGA-HF with the modulated CRET signal showed very relevant increases (70.7 ± 39.82%) over the controls in the expression of β-catenin. However, they were not statistically significant because the response showed a lot of interpersonal variability, with different responses depending on the donor. Regarding the standard signal, no relevant changes were observed compared to the control (Figure 6).

### 2.7. Effect of CRET on the Proportion of Bulge Cells of AGA-HF

One of the factors involved in the development of alopecia is the presence or absence of bulge cells and their ability to convert multipotent stem cells into cells of the hair follicle, sebaceous gland and epidermis. The absence of these cells produces degeneration of the HF, which gives rise to alopecia. To analyze the role of AGA-HF bulge cells in the potential effects of CRET, a specific marker of this cell type was used: cd-200. The results in the rate of cd-200+ cells showed a slight decrease compared to the control in their proportion with both signals, although it was not statistically significant (Figure 7).

### 2.8. Effect of CRET on the Proportion of Melanoblasts in AGA-HF

The differentiation of HF stem cells gives rise to various cell types of the follicle, among which are melanoblasts. As an indirect measure of the differentiation capacity of follicle stem cells after CRET treatment, the proportion of melanoblasts around the HF papilla was studied. For this, a specific marker was used, NKI/beted-gp100. The results showed a significant increase compared to the control in the number of NKI/beted-gp100+ cells present in the AGA-HF bulb matrix, after being treated with the standard signal. No significant effects were found in the samples treated with the modulated signal compared to their control (Figure 8).

## 3. Discussion

Currently, the development of new therapies for the treatment of AGA that present the minimum possible side effects and at the same time are effective is a growing demand among medical professionals and the population suffering from this pathology. RF-based therapies are a therapeutic alternative with few undesirable effects and promising results in clinical practice for the treatment of AGA. Thus, a preliminary study of 24 patients with AGA treated with a fractional RF device showed a favorable clinical response in 54% of cases, of which 40% experienced an increase of 30% or more in hair count, compared to the initial value [37]. Another clinical study used non-ablative fractionated RF devices as monotherapy to stimulate hair growth in 25 patients with AGA. The results demonstrated a 31.6% increase in hair density and an 18% increase in hair shaft thickness [15]. Although these clinical effects seem to indicate that this type of physical therapies can promote hair growth and maintenance in humans, clinical trials are very scarce and the mechanisms of such effects have not yet been fully described.

In this study, the potential ex vivo efficacy of CRET RF physical therapy in AGA was analyzed. To this end, the effect of treatment with CRET currents on the growth, death and differentiation of hair follicles from patients with this pathology was studied.

First, the effect of CRET on the proliferation and death by apoptosis of the different AGA-HF cell populations was analyzed using the Ki67 marker and TUNEL assay. Ki67 is associated with the proliferative phases of the cell cycle, as its expression appears during the G1 phase of the cycle, increases during the following phases, and decreases dramatically after mitosis. Ki67 has a very short half-life (no more than 1.5 h), so it does not accumulate in quiescent cells [38]. When AGA-HFs were treated with the standard signal, the proportion of Ki67+ cells increased significantly in the matrix, epidermis and bulge, while with the modulated signal, only the Ki67+ population in the sheath area increased. Furthermore, the standard signal also decreased sheath cell apoptosis. Thus, the standard CRET signal would be more efficient in increasing proliferation and inhibiting death in the different AGA-HF populations than the modulated signal.

The bulge cells are responsible for renewing the lower portion of the follicle in each anagen phase and therefore, all cell types in this region are derived from it. Furthermore, in case of disruption of the skin surface, the bulge cells migrate to the epidermis to contribute to tissue repair. On the other hand, the cells of the dermal sheath play an important role in the maintenance of the follicular papilla, since they act as a reservoir for its cells, while the cells of the matrix, through interaction with the follicular papilla, determine the thickness, length and cycle of the hair [39]. Therefore, a greater proliferation of these cells and less death induced by electric treatment would favor the renewal of the follicle and increase the reservoir of stem cells and specialized cells. Furthermore, the increase in the proliferation of the epidermis that surrounds the follicle would increase its depth, which would favor its stability. Together, these parameters would promote the strengthening of the follicle that would lead to less hair loss.

The extracellular matrix is an essential component of the follicle structure. MMPs are proteins responsible for hair renewal and play a very relevant role in hair regeneration. An increase in the degradation of the extracellular matrix leads to a weakening in the structural support of the hair follicle, which can contribute to miniaturization and eventual hair loss. HF exposures to exogenous physical factors such as UVB light (50 mJ/cm^2^) can cause an increase in the expression of MMPs induced by the generation of Reactive Oxygen Species (ROS), which affects the degradation and remodeling of the matrix and, as a consequence, its function [40]. Within the MMPs, MMP9 degrades collagen type I and collagen type II of the extracellular matrix [41]. Therefore, an inhibitory effect on its expression would prevent the degradation of the matrix and would generate a structural reinforcement in the morphology of the hair follicle, which would make hair loss more difficult. The present study has shown that CRET treatment provokes a response on MMP9 dependent on the type of signal used. Thus, treatment with the standard signal, but not with the modulated signal, caused a significant decrease in the rate of cells expressing MMP9. This effect would promote the structural maintenance of the follicle. It should be noted that such a decrease did not translate into a greater amount of total collagen or type I collagen in the treated AGA-HF.

Adult somatic stem cells act as the ultimate source of cells for self-renewing epithelia during homeostasis and wound healing. In the skin [42], the region of the bulge and the outermost layer of the outer root sheath of the HF contains a reservoir of quiescent stem cells (HFSCs). These stem cells show a high degree of plasticity [43] and upon appropriate stimuli are activated to drive the onset of anagen and begin a new hair cycle [44], leading to hair regeneration. The HFSCs of the bulge give rise to the different populations of the adult follicle, with keratinocytes being the most abundant in the HF. However, the relevance of the bulge from a clinical point of view is not limited only to keratinocytes, but also to the melanocyte stem cells resident in this area [45]. Indeed, it has been described that the follicle contains different subpopulations of follicular melanocytes, which are distinguished by their degree of differentiation and their ability to synthesize melanin [46,47]. Melanoblasts migrate along the outer root sheath (ORS) to reach the hair bulb around the dermal papilla (DP) [48]. In this compartment, due to the effect of various signals from surrounding cells (mainly from the DP), melanoblasts differentiate into melanogenically active melanocytes and will proceed with melanogenesis to generate pigmented hair shafts [5,48]. Melanogenesis in hair follicles is closely linked to the hair growth cycle and more precisely to the anagen phase [46]. Senescent HFSCs are cyclically eliminated from the skin through terminal epidermal differentiation [49]. When the supply of melanoblasts from the bulge is depleted, or an incomplete differentiation process occurs, loss of pigmentation of the hair shaft and the appearance of gray hair occur [50]. Given the relevance of bulge cells and melanoblasts in pathologies such as alopecia and graying, in the present study, the expression of specific markers of both cell types was studied. Although standard CRET did not induce significant changes in the expression of the specific marker of bulge stem cells (cd-200), it did increase the number of available melanoblasts, thus promoting melanogenesis in AGA-HF. Such an increase in melanoblasts would also indicate a promotion of CRET-induced differentiation of follicle stem cells towards more specialized cell types. This effect of CRET, together with the increase in the proliferation of different cell populations, would suggest a structural improvement of AGA-HF.

On the other hand, hair development involves an interaction between the epidermal and dermal compartments of the hair follicle [5,51]. Potential mechanisms involved in hair growth include, among others, the lengthening of the anagen phase, and therefore aged HFs usually show a reduced ability to enter this phase of hair growth [49]. It has been described that the activation of hair growth is carried out through the stimulation of the Wnt/β-catenin signaling pathway [52]. This signaling pathway is a regulator of cells that form the outer root sheath (ORS), matrix cells (DP) and dermal papilla cells, during hair morphogenesis and HF regeneration [52,53]. When this Wnt/β-cat pathway is stimulated, HFSCs are activated to become “transient amplification cells”, which proliferate, differentiate and transmit signals, thus initiating hair growth [50]. It has been proposed that this pathway could be a potential target in the treatment of hair loss, since its inhibition in AGA has been described [54]. Additionally, although a causal relationship between senescence and androgenic alopecia has not been demonstrated, there is a growing number of data that support that aging and AGA are closely linked, due to the inhibition of the Wnt/β-cat pathway and the DNA damage capable of causing HF cellular senescence, leading to its miniaturization [55]. In this study, the standard CRET signal did not cause changes in the expression of β-catenin compared to controls, although the modulated signal did slightly increase its expression. However, due to interpersonal variability, the results were not statistically significant.

The response described in AGA-HF involves an increase in proliferation and differentiation, as well as a reduction in death in the different cell types that comprise it. Previous studies revealed that the same CRET treatment parameters used in this study induced cell differentiation, migration and/or proliferative responses in different cell types, such as human stem cells [56,57], keratinocytes, fibroblasts [58] and dermal papilla cells (DPCs) [59], which are part of the hair follicle. These effects would be mediated by changes observed in the expression and location of β-catenin, E-cadherin, vinculin, p-FAK, MMP-9, and the MAP kinases (proteins that control both processes) p-p38, p-JUNK and p-ERK1/2. The effects described in the present work on proliferation, apoptosis and differentiation would induce the entry and permanence of hair follicles in anagen and in this way, electrical stimulation could favor hair growth and follicular proliferation. Furthermore, the increase in melanoblasts in the bulge, induced by CRET, could reduce the loss of color in the hair fiber and favor the differentiation of the different cell populations that compose it, which would lead to the strengthening of the hair fiber structure.

In parallel to this ex vivo study, our group carried out a pilot clinical trial with patients with androgenic alopecia in which the real usefulness of CRET therapy was analyzed, with promising results [59]. This clinical study was carried out in 20 patients with female pattern hair loss (FPHL) and were treated with the same 448 KHz CRET signal. Trichoscopic data revealed widespread and statistically significant hair redensification (10–15% over pre-treatment values) in all areas of the scalp. Given that the deregulation of HF cell viability and differentiation are the main factors underlying abnormal hair loss, the biological effects described here could be strongly involved in the redensifications observed in the trichological study mentioned above and would lead to a reduction in hair loss.

## 4. Materials and Methods

### 4.1. Human Hair Follicle Collection

One hundred and thirty human HF left over from hair graft surgeries were used. The follicles were kindly donated by Elite Laser Clinic (Madrid, Spain). These HF came from 6 patients between 30 and 45 years old, undergoing elective hair transplant surgery for AGA. This study was approved by the Ethic Committee of Ramón y Cajal Hospital and complied with the Declaration of Helsinki. All patients gave written informed consent before participating in this study. Hair follicles were transported to the laboratory in sterile culture tubes containing RPMI 1640 supplemented with 10% fetal bovine serum, 2 mM glutamine, penicillin 100 U/mL and streptomycin 100 µg/mL, which were placed in an ice container.

### 4.2. In Vitro Culture of Hair Follicles

Each isolated follicle was immediately placed into 60 mm plate filled with 4 mL of Williams medium E (Gibco™, Thermo Fisher Scientific, Waltham, MA, USA). In addition, the medium was supplemented with 2 mmol/L L-glutamine (Gibco, Thermo Fisher Scientific), hydrocortisone 10 ng/mL (Gibco, Thermo Fisher Scientific), insulin 10 μg/mL (Sigma-Aldrich, St. Louis, MO, USA), penicillin 100 U/mL (Gibco, Thermo Fisher Scientific), streptomycin 100 µg/mL (Gibco, Thermo Fisher Scientific), and amphotericin B 25 µg/mL (Gibco, Thermo Fisher Scientific). All HF were incubated at 37 °C in an atmosphere of 5% CO_2_ and 95% air.

### 4.3. Electric Treatment

Pairs of sterile stainless-steel electrodes designed ad hoc for in vitro stimulation were inserted in all Petri dishes and connected in series. Only the electrodes of dishes for electrical stimulation were energized using a signal generator (Indiba Activ HCR 902, INDIBA^®^, Barcelona, Spain), while the remaining plates were sham-exposed simultaneously in identical conditions. The follicles were treated with two different types of signals: a standard 448 kHz sinusoidal, non-modulated signal (CRET-Std) or 20 kHz, 40% amplitude modulation of the 448 kHz signal (CRET-Mod). In both cases, electrical stimulation was applied intermittently. The intermittent stimulation pattern consisted of 5 min pulses of 448 kHz, sine wave currents delivered at subthermal densities of 100 µA/mm^2^, separated by 4 h interpulse lapses, and administered for a total of 48 h.

### 4.4. TUNEL Assay

TUNEL assay was evaluated using the Apoptosis Detection System (Promega, Madison, WI, USA). For TUNEL staining of tissue samples, formalin-fixed, paraffin-embedded (FFPE) tissue blocks were sectioned with a thickness of 3 µm and dried for 15 min at 60 °C, before being dewaxed in xylene, rehydrated by a graded ethanol series, and washed with phosphate-buffered saline (PBS). The apoptotic index was expressed as a percentage of TUNEL-positive cells/total cells.

### 4.5. Immunohistochemistry for Ki67, β-Catenin, MMP9, Collagen Type I, NKI/Beted and cd-200

HF sections (3 μm) were deparaffinized and rehydrated as described previously [60] and antigen retrieval was achieved by heat treatment in a pressure cooker for 2 min in 10 mM citrate buffer (pH 6.5). Next, endogenous peroxidase was blocked, and the sections were incubated with anti Ki67 (1:1000) (Abcam, Cambridge, UK, ab16667), β-catenin (1:200) (Santa Cruz Biotechnology, Dallas, TX, USA, sc-7963), Metalloproteinase 9 (1:1000) (Abcam 76006-1), NKI/beted (1:100) (Monosan, Uden, The Netherlands, Mon7006-1), cd-200 (1:5000) (Biorad, Munich, Germany, MCA 1960GA); Collagen I (1:1000) (Sigma SAB4200678) overnight at room temperature. The following day, sections were incubated with the Master Polymer Plus Detection System (Persoxidase) (MAD-000237QK-10) (Master Diagnostica, Granada, Spain), according to the manufacturer’s instructions. The sections were counterstained with hematoxylin.

### 4.6. Hematoxylin and Masson’s Trichrome Staining for Total Collagen

Hematoxylin staining is the gold standard for microscopic observation and diagnosis. However, other additional stains are used for specific purposes, such as trichrome stains for staining collagens in the extracellular matrix. Both types of staining were used for the morphometric study and the collagen content assessment of the extracellular matrix of hair follicles. The HFs were fixated in 10% neutral buffered formalin, embedded in paraffin wax. The HFs sections were routinely dewaxed, rehydrated and washed with destilled water and stained with hematoxylin or Masson’s trichrome staining (for total collagen) for histopathological evaluation.

### 4.7. Histomorphometric Analysis and Assessment of Immunohistochemistry Images

Microscopy images were taken with different magnification for histomorphometric analysis and immune characterization. Microscopic captures of cut samples were taken with a digital microscope (Nikon-Eclipse-T2R, Nikon Instruments Inc., Melville, NY, USA). At least 10 random fields for each slide were analyzed. Image-J analysis was used to measure the different parameters. The expression of each of the markers was expressed as a percentage of positive cells/total cells and normalized over the control. The epidermis depth measurement was evaluated by taking at least five measurements in each of the images in both experimental groups. At least 15 images were taken per experimental group.

### 4.8. Statistic Analysis

All experimental procedures and analysis were conducted blindly for treatment. Statistical analyses were performed with Graph-Pad Prism 6.01 software (GraphPad Software, Inc., La Jolla, CA, USA). Data were normalized and expressed as means ± standard error (SEM). The two-tailed Student’s *t*-test was used. The limit of statistical significance was set at *p* < 0.05.

## 5. Conclusions

The present study shows the potential efficacy of CRET treatment for the treatment of AGA. Thus, the application of the CRET signal in follicles from patients with this pathology increases the proliferation and reduces the death of different AGA-HF cell populations. In addition, it induces strengthening and could reduce the loss of color in the hair fiber. These effects induced by CRET would be compatible with a prolongation of or re-entry into the anagen phase of the hair cycle and would lead to a reduction in the loss of hair follicles affected by AGA. Taken together, these results would support the effectiveness of RF-based therapies for the treatment of alopecia. However, these data come from an ex vivo study, which is a limitation for direct extrapolation to patients. Therefore, it will be necessary to carry out clinical trials to know the true effectiveness of CRET therapy and other RF therapies for the treatment of AGA.

## Figures and Tables

**Figure 1 ijms-25-07865-f001:**
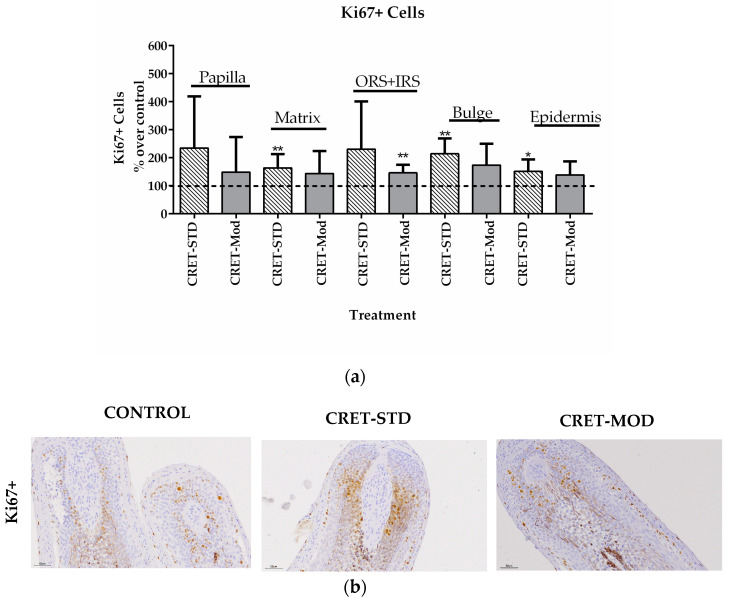
Effect of CRET on AGA-HF proliferation. (**a**) Rate of Ki67+ cells in the different areas of the follicle. Percentage of effect over the control in follicles treated with the standard or modulated signal. All data are presented as mean ± SEM and were analyzed with Student’s *t*-test. *, *p* < 0.05; **, *p* ≤ 0.001. (**b**) Representative images of control, standard CRET (CRET-Std) or CRET-modulated (CRET-Mod) follicles. Top: Ki67+ of the matrix zone of AGA-HG; below; Ki67+ of the AGA-HF sheath (ORS+IRS). Violet: Nuclei stained with hematoxylin. Brown: Ki67+ labeled with DAB. 200X magnification. Objective 20X.

**Figure 2 ijms-25-07865-f002:**
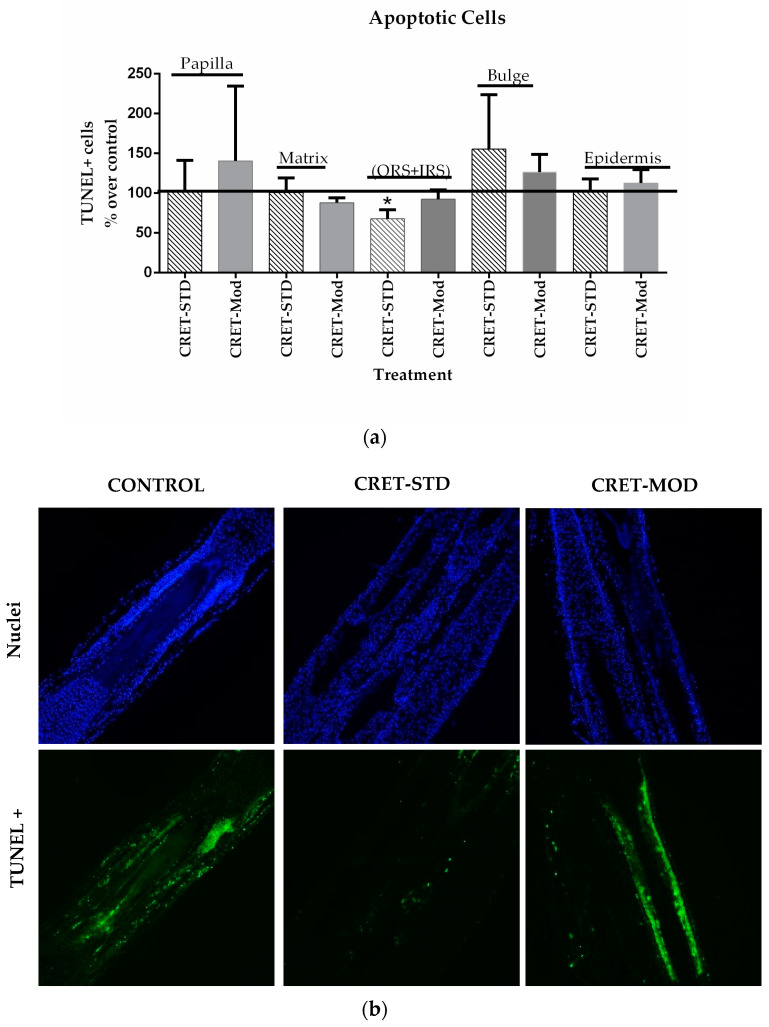
Effect of CRET on AGA-HF cell death. (**a**) Rate of TUNEL+ cells in the different areas of the follicle. Percentage effect on control TUNEL+ cells treated with the standard or modulated signal. All data are presented as mean ± SEM and were analyzed with Student’s *t*-test. *, *p* < 0.05. (**b**) Representative images of control, standard CRET (CRET-Std) or CRET-modulated (CRET-Mod) follicles. The images show only the TUNEL+ cells in the sheath area (IRS+ORS). Green: TUNEL+ cells; Blue: total nuclei. 200X magnification. Objective 20X.

**Figure 3 ijms-25-07865-f003:**
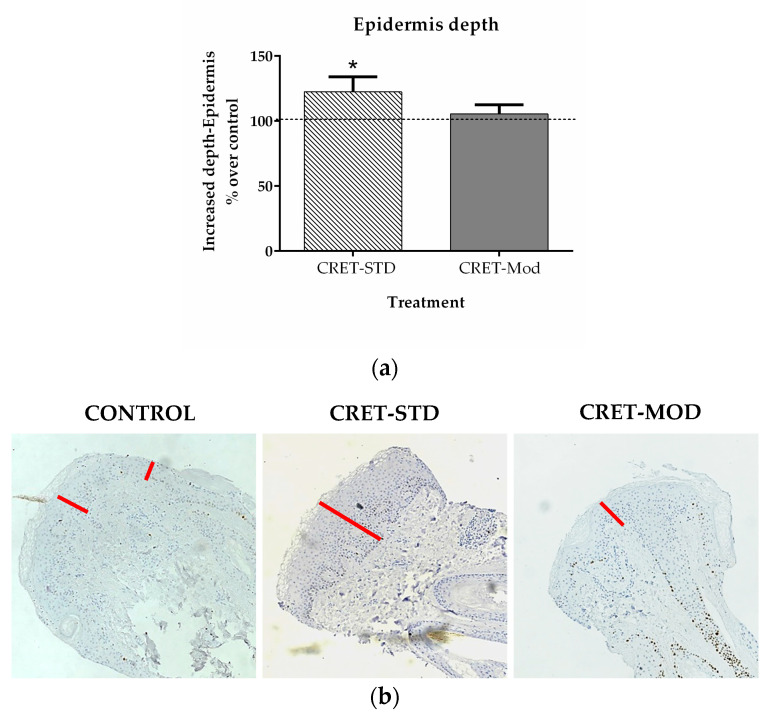
Effect of CRET on the depth of the AGA-HF epidermis. (**a**) Measurement of depth of the epidermis using the Image-J program. All data are presented as mean ± SEM and were analyzed with Student’s *t*-test. *, *p* < 0.05. (**b**) Representative images of control, standard CRET (CRET-Std) or CRET-modulated (CRET-Mod) follicles. At least 5 measurements were taken in the area of the epidermis of the different follicles. Red line: epidermis thickness. 200X magnification. Objective 20X.

**Figure 4 ijms-25-07865-f004:**
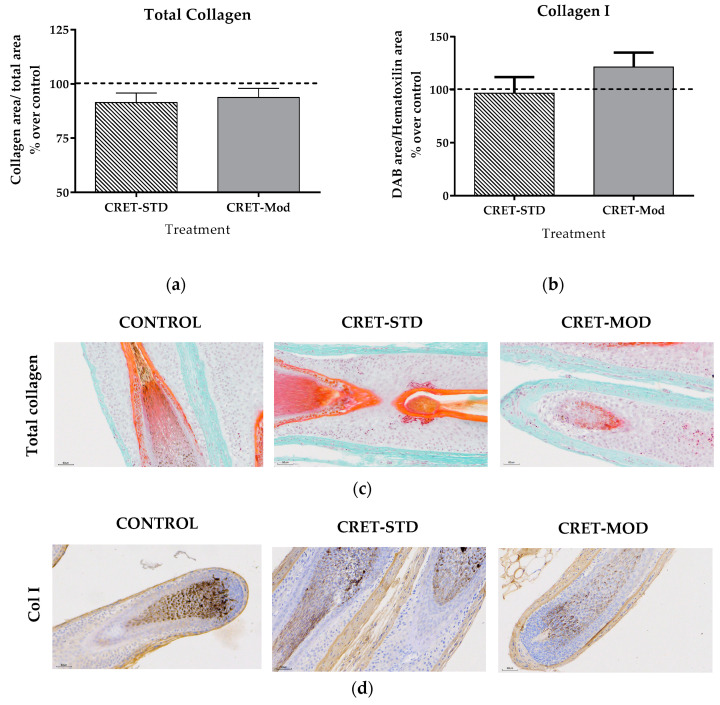
Effect of CRET on AGA-HF collagen expression. (**a**) Color measurement by Mason’s trichrome staining for quantification of total collagen. (**b**) Collagen I expression, measured as the ratio between the DAB area in relation to the hematoxylin area. All data are presented as mean ± SEM and were analyzed with Student’s *t*-test. NS, *p* > 0.05. (**c**) Representative images of Mason’s trichrome staining of the different areas of the follicles; control, standard CRET (CRET-Std) or CRET-modulated (CRET-Mod). Violet: Nuclei stained with hematoxylin; green: collagen; (**d**) Representative images of the immunohistochemistry for type I collagen of the different areas of the follicles; control, standard CRET (CRET-Std) or CRET-modulated (CRET-Mod). Violet: Nuclei stained with hematoxylin. Brown: type I collagen labeled with DAB. 200X magnification. Objective: 20X.

**Figure 5 ijms-25-07865-f005:**
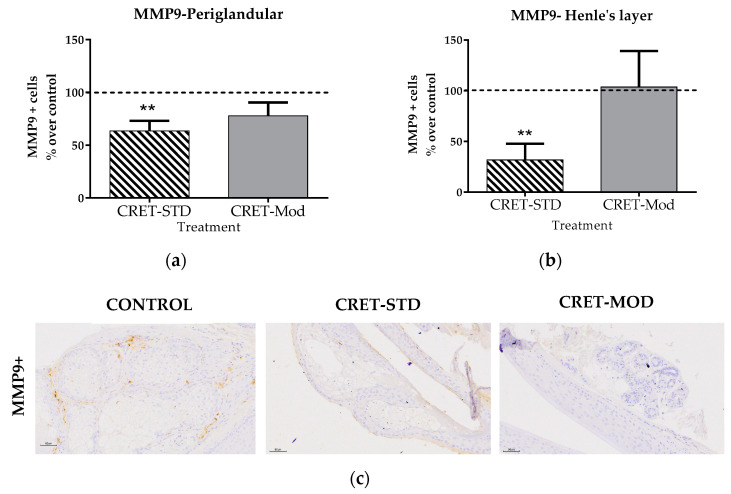
Effect of CRET on MMP9 expression of AGA-HF. (**a**) Rate of MMP9+ cells in the periglandular zone and (**b**) rate of MMP9+ cells in the layer of Henle zone. Percentage effect on control of MMP9+ cells treated with the standard or modulated signal. All data are presented as mean ± SEM and were analyzed with Student’s *t*-test. **, *p* ≤ 0.01. (**c**) Representative images of control, standard CRET (CRET-Std) or CRET-modulated (CRET-Mod) follicles. Violet: Nuclei stained with hematoxylin; brown: MMP9+ labeled with DAB. 200X magnification. Objective: 20X.

**Figure 6 ijms-25-07865-f006:**
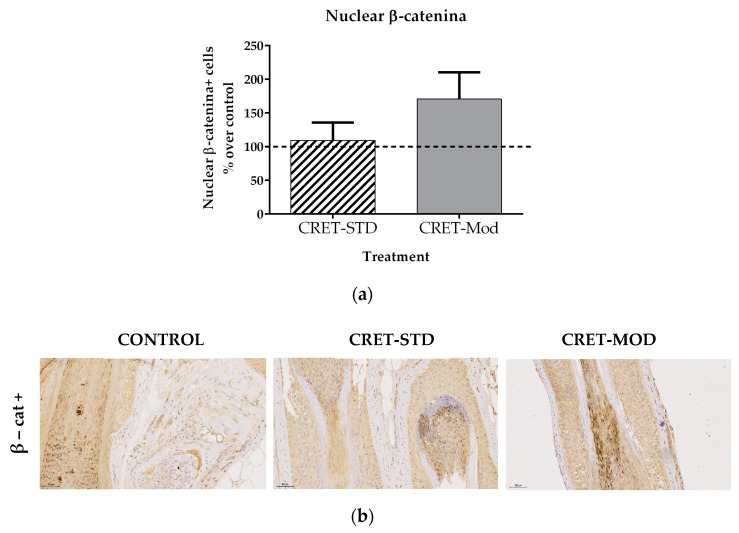
Effect of CRET on AGA-HF β-catenin expression. (**a**) Rate of β-catenin + cells. Percentage effect on control of β-catenin cells treated with the standard or modulated signal. All data are presented as mean ± SEM and were analyzed with Student’s *t*-test. (**b**) Representative images of control, standard CRET (CRET-Std) or CRET-modulated (CRET-Mod) follicles. Violet: Nuclei stained with hematoxylin. Brown: β-cat+ cells labeled with DAB. 200X magnification. Objective: 20X.

**Figure 7 ijms-25-07865-f007:**
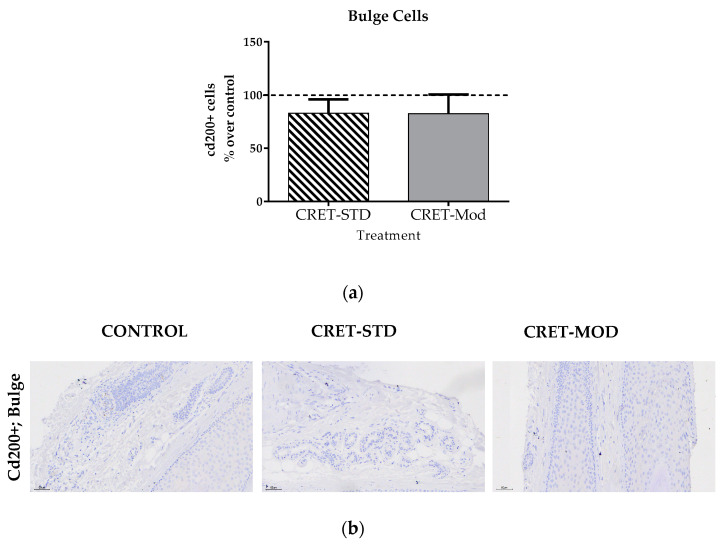
Effect of CRET on the number of AGA-HF bulge cells. (**a**) Rate of cd-200 + cells. Percentage effect on control of cd-200+ cells treated with the standard or modulated signal. All data are presented as mean ± SEM and were analyzed with Student’s *t*-test. (**b**) Representative images of control, standard CRET (CRET-Std) or CRET-modulated (CRET-Mod) follicles. Violet: Nuclei stained with hematoxylin. Brown: cd200+ cells labeled with DAB. 200X magnification. Objective: 20X.

**Figure 8 ijms-25-07865-f008:**
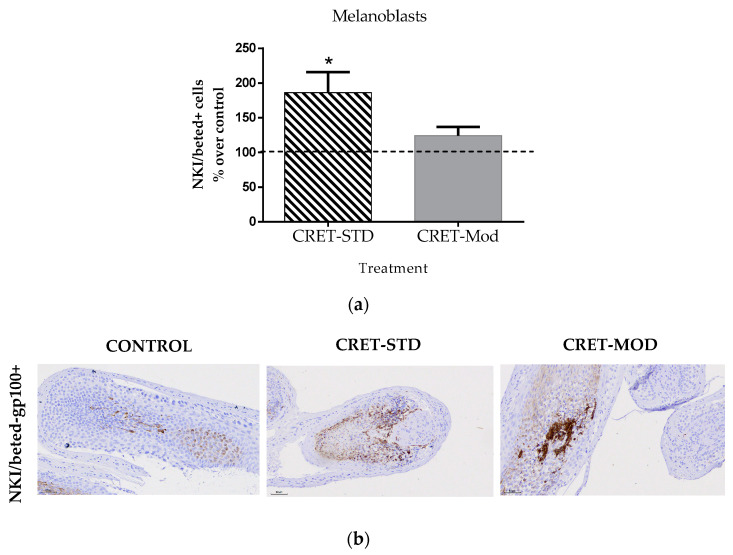
Effect of CRET on the number of AGA-HF melanoblasts. (**a**) Rate of NKI/beted-gp100+ cells. Percentage effect on control of NKI/beted-gp100+ cells treated with the standard or modulated signal. All data are presented as mean ± SEM and were analyzed with Student’s *t*-test. *, *p* < 0.05. (**b**) Representative images of control, standard CRET (CRET-Std) or CRET-modulated (CRET-Mod) follicles. Violet: Nuclei stained with hematoxylin. Brown: NKI/beted-gp100+ labeled with DAB. 200X magnification. Objective: 20X.

## Data Availability

The data presented in this study are available upon request from the corresponding author. The data are not publicly available due to privacy restrictions.

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
