# Peer review of "Effects of RF Electric Currents on Hair Follicle Growth and Differentiation: A Possible Treatment for Alopecia"

_ijms, 2024, doi:10.3390/ijms25147865_

Round 1

Reviewer 1 Report

Comments and Suggestions for Authors

The incidence of AGA is increasing, in female patients also. Therefore, any new possible treatment for AGA it is salutary. Hereby please find my comments regarding the paper:

1.     The abstract is well structured and the information’s are well presented.

2.     In introduction, please extend the data about other therapy used for AGA (ex. intralesional corticoids, cryotherapy).

3.     Please insert a subchapter that differentiate the use of RF on hair follicle growth and differentiation versus RF to hair removal.

4.     The results and the discussions are clearly exposed.

5.     The conclusions summarized the effect of RF on ex-vivo hair and draw future lines of research.

6.     The references are appropriate.

7.     The quality of figures and the data are good.

Author Response

Thank you for your review of our study and for your helpful comments. They will surely help to improve our study and that readers can better understand the results. Below, we discuss the changes made to the manuscript following their comments.

The incidence of AGA is increasing, in female patients also. Therefore, any new possible treatment for AGA it is salutary. Hereby please find my comments regarding the paper:

  1. The abstract is well structured and the information’s are well presented.
  2. In introduction, please extend the data about other therapy used for AGA (ex. intralesional corticoids, cryotherapy).

Response 2: We have added new information in the introduction of the manuscript in which the latest findings in these two therapies for the treatment of alopecia are discussed. Additionally, we have added updated bibliography on the topic.

  1. Please insert a subchapter that differentiate the use of RF on hair follicle growth and differentiation versus RF to hair removal.

Response 3: We have added a new paragraph commenting on the difference between both radiofrequency therapeutic modalities. In addition, we have added updated bibliography on the topic.

  1. The results and the discussions are clearly exposed.
  2. The conclusions summarized the effect of RF on ex-vivo hair and draw future lines of research.
  3. The references are appropriate.
  4. The quality of figures and the data are good.

Reviewer 2 Report

Comments and Suggestions for Authors

 An interesting paper is discussed, containing a study on the therapeutic potential of CRET therapy in the treatment of androgenic alopecia. However, there are few very critical questions and concerns required to be answered for strengthening the validity and impact of the manuscript. An underlying biological mechanism of maximum essence is how CRET therapy affects hair follicle proliferation and apoptosis. The study somewhat raises this point in quite vagueness. How are the electrical currents specifically interacting with the cellular components of the hair follicle for the observed change in result? Could more proof or references be given to support the proposed mechanisms?  

The study adopts an ex-vivo model to investigate the impact of CRET treatment, thereby eliciting a gap in translation of these results to in vivo conditions. What limitations may be present for using ex-vivo hair follicles to predict clinical results? How can you ensure that the effects observed will be replicated in real patients suffering from AGA? Moreover, with the variability in hair follicles from different donors, how do you minimize variability and standardize the sample?  

In the Results section, a number of the measures cited reference statistical significance; however, the biological significance must be more fully elaborated. For example, changes in Ki67+ cells and TUNEL+ cells are statistically significant, but what does this actually mean in regard to the general health and functioning of the hair follicle? Do such alterations on the cellular level actually translate into alterations on the macro-levels of hair density, thickness, or rate of growth? Although this manuscript is written well, there are places within it that could be clearer, with more consistency in the terminology used. Checking to make sure all terms are clearly defined and then following those definitions in the document can help ensure consistency in the text.  

Discussion of clinical relevance with CRET therapy is interesting yet somewhat speculative, and because this is an ex-vivo study, to what extent can one conclude that increases in proliferation and decreases in apoptosis are likely to have a significant clinical benefit in terms of hair regrowth or loss prevention? What are the next steps to transition from ex-vivo findings to clinical trials? Further, interpersonal variability in the response to CRET therapy is discussed. How large would this variation be, and what factors could possibly give rise to such diverging responses across people?  

Essentially, it is a good study with important preliminary data for CRET therapy in AGA. However, these critical questions and concerns will definitely be very important and help make the manuscript more impactful and valid, and worthwhile for a high-impact journal.

Comments on the Quality of English Language

Minor editing of English language required

Author Response

Thank you for your review of our study and for your helpful comments. They will surely help to improve our study and that readers can better understand the results. Below, we answer your questions point by point.

Additionally, we have made changes to the manuscript following your comments. We have also introduced new bibliography.

Comments 1: An underlying biological mechanism of maximum essence is how CRET therapy affects hair follicle proliferation and apoptosis. The study somewhat raises this point in quite vagueness. How are the electrical currents specifically interacting with the cellular components of the hair follicle for the observed change in result? Could more proof or references be given to support the proposed mechanisms? 

Response 1: To analyze how electrical currents interact specifically with the cellular components of the hair follicle, a manuscript has also been presented for this same special issue that shows the in vitro effect of CRET (standard or modulated) on primary cultures of dermal papilla cells (DPC). This manuscript is under review. The objective of our in vitro study was to analyze, for the first time, the ability of the CRET electrical signal to induce effects in DPC, a region of the follicle directly involved in hair growth. The intention of this study was to characterize, through a preclinical in vitro assay, basic biological processes such as proliferation, cell death and differentiation. To study these processes, specific markers capable of supporting a possible effect were used. For this purpose, proliferation assays such as XTT were performed, as well as markers involved in proliferation such as Ki67 and ERK1/2. For cell death assays, p53 and caspase 3 expression analysis was analyzed, and hair follicle cell differentiation was performed by versican and PPARγ expression analysis. On the other hand, although not all response pathways have been covered, it has been possible to obtain a first approximation of the possible pathways involved in the effect of the electrical signals used on DPC. We add attached a diagram with the possible pathways involved in the different effects triggered by the different signals used in these DPC cultures.

Other previous studies also revealed that the same exposure parameters used in this study induced proliferative and cell differentiation responses in different cell types such as human stem cells, keratinocytes and fibroblasts, some of which, such as stem cells or keratinocytes, are part of the hair follicle. These articles also discuss potential molecular pathways to CRET. These articles also discuss potential molecular pathways to CRET in these cell types. (In vitro stimulation with radiofrequency currents promotes proliferation and migration in human keratinocytes and fibroblasts July 2021. Electromagnetic Biology and Medicine 40(3):1-15 DOI: 10.1080/15368378.2021.1938113; Chondrogenic Differentiation of Adipose-Derived Stem Cells by Radiofrequency Electric Stimulation January 2017 Journal of Stem Cell Research & Therapy 7(12) DOI: 10.4172/2157-7633.1000407; Electric Stimulation at 448 kHz Promotes Proliferation of Human Mesenchymal Stem Cells November 2014Cellular Physiology and Biochemistry 34(5):1741-1755 DOI: 10.1159/000366375). 

Comments 2: The study adopts an ex-vivo model to investigate the impact of CRET treatment, thereby eliciting a gap in translation of these results to in vivo conditions. What limitations may be present for using ex-vivo hair follicles to predict clinical results? How can you ensure that the effects observed will be replicated in real patients suffering from AGA? Moreover, with the variability in hair follicles from different donors, how do you minimize variability and standardize the sample?  

Response 2: Indeed, these data come from an ex vivo study, which represents a limitation for direct extrapolation to patients. This is because ex vivo systems do not consider all the factors involved in the in vivo response and therefore, the conclusion that can be reached from their application is limited. Therefore, in parallel to this ex vivo study, our group carried out a pilot clinical trial with patients with androgenic alopecia in which the real usefulness of CRET therapy was analyzed, with promising results. This clinical study was carried out on 20 patients with female pattern hair loss (FPHL) and they were treated with a 448 KHz CRET signal. This study was published in Pablo N, et al. Radiofrequency Current at 448 Khz For Female Pattern Hair Loss: Cellular Bases For Redensification Improvement. J Dermatol Res. 2022;3(2):1-24. DOI: https://doi.org/10.46889/JDR.2022.3209. The study revealed that the treatment caused a statistically significant capillary redensification of between 10 and 15% compared to the previous treatment in all treated areas. We believe that these three studies (in vitro, ex vivo and pilot clinical trial) complement each other to give a broader view of the effect of CRET on hair regeneration.

Regarding how to minimize variability and standardize the sample, it must be taken into account that biological samples from different individuals present an intrinsic variability that is very difficult to eliminate, due to factors such as the individual's own genetic factors. Alopecia disease is complex. The cause can be genetic or acquired or secondary to other conditions. As a consequence, the response to treatment will not always be of the same intensity. However, several options can be proposed to try to reduce this variability. One option would be to obtain follicles from donors of very similar ages and clinical conditions, which would reduce this variability and standardize the sample. Another option would be to increase the number of donors, thus reducing the variability of results, which would allow stratification by groups based on the clinical characteristics of the donors.

Comment 3: In the Results section, a number of the measures cited reference statistical significance; however, the biological significance must be more fully elaborated. For example, changes in Ki67+ cells and TUNEL+ cells are statistically significant, but what does this actually mean in regard to the general health and functioning of the hair follicle? Do such alterations on the cellular level actually translate into alterations on the macro-levels of hair density, thickness, or rate of growth? Although this manuscript is written well, there are places within it that could be clearer, with more consistency in the terminology used. Checking to make sure all terms are clearly defined and then following those definitions in the document can help ensure consistency in the text.  

Response 3: The main mechanism underlying abnormal hair loss is the degradation of hair follicles due to their miniaturization. This miniaturization is caused by a deregulation of the hair growth phases, which may be due to a prolongation of the catagen, exogen or kenogen phases, or a delay or shortening of the anagen phase.

The proliferative, differentiation and anti-apoptotic response of FH involves increased proliferation and differentiation as well as reduced death of the different cell types that comprise it. This would induce the entry and permanence of hair follicles in anagen and in this way, electrical stimulation could favor hair growth and follicular proliferation. These biological effects could be strongly involved in some of the redensifying effects observed in the trichological study. This electroinduced effect could be added to those of the thermal and mechanical stimuli of the CRET treatment, whose potential contribution to the observed hair redensification has not yet been sufficiently characterized or quantified.

Comment 4: Discussion of clinical relevance with CRET therapy is interesting yet somewhat speculative, and because this is an ex-vivo study, to what extent can one conclude that increases in proliferation and decreases in apoptosis are likely to have a significant clinical benefit in terms of hair regrowth or loss prevention? What are the next steps to transition from ex-vivo findings to clinical trials? Further, interpersonal variability in the response to CRET therapy is discussed. How large would this variation be, and what factors could possibly give rise to such diverging responses across people?  

Response 4: Although it is not possible to extrapolate the results of the ex vivo study, results of the clinical study show that three months after the end of the CRET treatment, the patients presented, as a whole, a significant increase in capillary redensification, as revealed by the analysis of the trichoscopic data in the derived Sinclair scale. This effect is due to the significant changes recorded in a variety of parameters involved in hair regeneration, including increases in the average number of hairs and in the number of follicular units. Such changes were detected both in areas with deficient capillary density according to the pre-treatment diagnosis, and in those whose capillary density had been considered normal. Furthermore, it revealed significant increases in the rate of triple follicular units, at the cost of corresponding reductions in the rates of single and double follicular units. This is indicative of the potential ability of RF electrothermal treatment to increase the diameter of follicular subunits that had been miniaturized by alopecia.

Regarding interpersonal variation, it was also observed in the clinical trial carried out and mentioned above. In this clinical study, analysis of individual responses also revealed notable differences between patients in the level of response to treatment. In fact, while some patients showed improvements in several redensification parameters and in several treated areas, other patients showed improvements in fewer parameters and/or in fewer areas. Analysis of the patients' records and histories revealed no significant relationships between their degree of response to treatment and factors such as age, hair follicle characteristics, scalp characteristics, and previous diseases. Therefore, we do not know the cause of these interpersonal variations.

Round 2

Reviewer 1 Report

Comments and Suggestions for Authors

Dear authors,

The suggested changes are made and increased the value of the paper.

Reviewer 2 Report

Comments and Suggestions for Authors

Thank you for your responses.